# Uncovering the Epitranscriptome: A Review on mRNA Modifications and Emerging Frontiers

**DOI:** 10.3390/genes16080951

**Published:** 2025-08-12

**Authors:** Douglas M. Ruden

**Affiliations:** Department of Obstetrics and Gynecology, C. S. Mott Center for Human Growth and Development, Institute of Environmental Health Sciences, Wayne State University, Detroit, MI 48201, USA; douglasr@wayne.edu

**Keywords:** mRNA modification, epitranscriptomics, nanopore sequencing, pseudouridine, methyladenosine, environmental mRNA, mRNA caps

## Abstract

**Background/Objectives:** Messenger RNA (mRNA) modifications regulate key steps in gene expression, including splicing, translation, and stability. Despite over 300 known RNA modifications, the relatively small subset occurring in mRNA remains understudied compared with tRNA and rRNA. This review aims to systematically evaluate 15 known naturally occurring mRNA-specific modifications, rank them by publication frequency, and highlight emerging frontiers in epitranscriptomics, including discovering new naturally occurring mRNA modifications and environmental RNA (eRNA) epitranscriptomics. **Methods:** We conducted a structured literature review of PubMed-indexed publications to rank mRNA modifications by citation prevalence. Key modifications such as m^6^A, m^5^C, Ψ, and m^1^A were analyzed in terms of enzymatic machinery (“writers,” “erasers,” and “readers”), molecular functions, and physiological relevance. We also reviewed technological advances, with a focus on nanopore sequencing for detection of RNA modifications in native and environmental contexts. **Results:** The modification m^6^A was identified as the most studied mRNA modification, followed by Ψ, m^5^C, and A-to-I editing (inosine). These modifications influence diverse mRNA processes, including translation efficiency, localization, and immune evasion. Cap-specific modifications such as Cap0, Cap1, and Cap2 were also described, highlighting their role in transcript stability and innate immune regulation. Advances in nanopore sequencing have enabled direct detection of RNA modifications and offer promise for eRNA (environmental RNA) surveys. The potential for nanopore sequencing of many other of the 335 known RNA modifications in the MODOMICS database using existing nanopore technologies is also discussed. **Conclusions:** mRNA modifications represent a critical, yet incompletely mapped, layer of gene regulation. Continued research—especially using nanopore and machine learning technologies—will help uncover their full biological significance. Exploration of eRNA and identifying new mRNA modifications will redefine our understanding of RNA biology.

## 1. Introduction

The regulation of gene expression extends far beyond the DNA sequence, encompassing a complex landscape of chemical modifications that affect RNA molecules. Among these, modifications of messenger RNA (mRNA)—the direct templates for protein synthesis—represent a rapidly emerging frontier in molecular biology. These chemical marks, collectively referred to as the epitranscriptome, play vital roles in mRNA processing, stability, translation, and localization [1,2,3].

Over 300 types of RNA modifications have been cataloged to date, with the majority studied in transfer RNA (tRNA) and ribosomal RNA (rRNA) (see MODOMICS database, [4]). However, only a subset of these modifications occurs on mRNA, where they are less understood but potentially more dynamic and responsive to cellular context. Prominent among these is N6-methyladenosine (m^6^A), which has become a paradigm for reversible, regulatory RNA modifications [5], which can now be performed at the single-cell level [6,7]. Other notable mRNA modifications include pseudouridine (Ψ or Y) [8,9], 5-methylcytidine (m^5^C) [8,10,11], N1-methyladenosine (m^1^A) [12,13], inosine (I) [14,15], and various mRNA cap modifications such as m^7^G [16,17] and m^6^Am [18,19] at the 5′ end [20].

The functional consequences of these modifications are profound: they can influence transcript lifespan, translation efficiency, splice site selection, and cellular localization. Moreover, several modifications help mRNAs evade innate immune sensors such as RIG-I, highlighting their role in immune regulation and therapeutic mRNA design [21,22]. RIG-I (retinoic acid-inducible gene I) is a crucial cytoplasmic RNA sensor that plays a vital role in the innate immune system’s response to viral infections. It functions as a pattern recognition receptor (PRR) that detects viral RNA and initiates a signaling cascade leading to the production of type I interferons, which are critical for antiviral defense. RIG-I is particularly important for recognizing RNA viruses, such as influenza and hepatitis C, by binding to specific viral RNA structures such as 5′-triphosphorylated RNA [21,22].

Despite rapid advances, key controversies persist. For example, while m^6^A is known to be dynamically regulated, there is debate over how reversible and site-specific its deposition truly is [13,14]. Similarly, the functional roles of modifications such as m^5^C and Ψ in mRNA remain incompletely defined and sometimes contradictory depending on cell type and experimental model [15]. The contradictory results might be explained by mRNA modifications like m^5^C and Ψ being involved in a wide range of cellular processes, including:RNA export: m^5^C can play a role in transporting mRNA from the nucleus to the cytoplasm.Translation: Both m^5^C and Ψ can influence the rate and fidelity of protein synthesis, potentially even leading to alternative protein products.mRNA stability: Ψ, for example, can enhance mRNA stability by affecting its structure and protecting it from degradation.Development and disease: Alterations in these modifications are linked to various physiological and pathological processes, including embryonic development and tumor formation.

The development of new sequencing technologies has revolutionized the field. Mass-spectrometry-based methods and antibody-enrichment sequencing techniques such as MeRIP-seq (methyl RNA immunoprecipitation) [23] and miCLIP (mRNA individual nucleotide resolution crosslinking immunoprecipitation) [24] have mapped many modifications, albeit with limitations in resolution and specificity [16]. More recently, nanopore sequencing has enabled direct RNA sequencing and detection of modifications in native RNA molecules without prior conversion to cDNA, opening new avenues for studying both endogenous and environmental RNA (eRNA) [17,18].

The aim of this review is to systematically summarize the major known modifications of mRNA, ranking them by their prevalence in the scientific literature as a proxy for research emphasis. We describe their biochemical mechanisms, molecular functions, and emerging detection technologies. A final section highlights future directions, including nanopore-based discovery of novel modifications in environmental contexts.

## 2. Materials and Methods

To rank mRNA modifications by their prevalence in the literature, we systematically queried the PubMed database (https://pubmed.ncbi.nlm.nih.gov) on 20 June 2025 using both the full chemical name and the abbreviated code for each RNA modification (e.g., “N1-methyladenosine” and “m^1^A”). Each modification listed in the MODOMICS database (https://iimcb.genesilico.pl/modomics/) on 20 June 2025 was included if it was classified as naturally occurring in cellular RNA. These included modifications found in messenger RNA (mRNA), transfer RNA (tRNA), ribosomal RNA (rRNA), and noncoding RNAs (ncRNAs).

To ensure the specificity of our results, search terms were restricted to exact phrase matches when possible, and abstracts were manually screened to confirm that the citation referenced RNA modifications rather than unrelated chemical analogs or non-RNA-related methylation. Both nucleoside (base + sugar) and nucleotide (base + sugar + phosphate) forms were included in the initial screening; however, final analysis focused exclusively on naturally occurring nucleosides.

The number of PubMed-indexed publications for each RNA modification was tallied and used as a proxy for research attention and biological relevance. These citation counts were used to rank the most studied mRNA modifications, and the results are presented. 

## 3. Results

### 3.1. Ranking of mRNA Modifications by PubMed Prevalence and Research Emphasis

To provide a contextual overview of the landscape of naturally occurring mRNA modifications, we systematically ranked them based on scientific attention as measured by the number of PubMed-indexed articles referencing each modification. Our approach aimed to capture relative research emphasis across different types of RNA chemical modifications, using this metric as a proxy for scientific visibility, biomedical interest, and historical momentum in the field.

The ranking process involved structured queries for each modification, incorporating both IUPAC chemical names and widely used abbreviations (e.g., “N6-methyladenosine” and “m^6^A”). Results were manually curated to remove irrelevant records, such as those exclusively focused on tRNA, rRNA, or DNA methylation, thereby increasing the specificity to mRNA-focused studies. For example, a search for “pseudouridine” was refined to include only articles that specifically investigated Ψ in mRNA contexts, which excluded many Ψ studies in tRNA.

While PubMed is the most widely used and standardized biomedical literature database, we acknowledge the limitations of relying solely on PubMed citation counts as a measure of biological or functional relevance. Certain modifications may be underrepresented because of technical challenges in detection (e.g., lack of selective antibodies or chemical derivatization methods), despite their likely physiological importance. Conversely, more “popular” modifications such as m^6^A or m^5^C have benefited from early availability of mapping tools, commercial reagent development, and their association with high-profile disease mechanisms such as cancer and neurodegeneration. As such, this ranking reflects not only biological prevalence but sociotechnical factors including method accessibility, commercial tool development, and translational interest.

In summary, while PubMed citation counts provide a useful baseline for understanding how the field has historically prioritized different RNA modifications, they must be interpreted within a broader framework that considers technical limitations, emerging technologies, and the evolving conceptual landscape of epitranscriptomic regulation. These rankings, listed in Table 1, should thus be viewed as a living snapshot of research activity rather than a definitive statement of biological relevance.

#### 3.1.1. N6-Methyladenosine (m^6^A): A Central Regulatory Node

N6-methyladenosine (m^6^A) is the most extensively studied mRNA modification, with over 7000 publications indexed in PubMed (Figure 1). This modification is dynamically regulated by the METTL3–METTL14 methyltransferase complex, which installs the mark, and by the demethylases FTO and ALKBH5, which remove it. Reader proteins such as YTHDF1–3 and YTHDC1 interpret m^6^A marks to influence mRNA fate. m^6^A is typically enriched in the 3′ untranslated region (UTR) and near stop codons, where it promotes regulated mRNA decay, translational control, alternative splicing, and nuclear export.

Functionally, m^6^A marks have been shown to fine-tune transcript dosage in response to developmental and environmental cues. In embryonic stem cells, m^6^A destabilizes pluripotency factors to drive differentiation. Under stress conditions, m^6^A enhances translation of heat-shock proteins. Circadian rhythm, synaptic plasticity, and antiviral responses also depend on m^6^A-mediated regulation. Dysregulation of m^6^A is implicated in several diseases: METTL3 overexpression in glioblastoma promotes tumorigenicity by stabilizing oncogenic transcripts, while FTO upregulation contributes to obesity and chemoresistance in leukemia. The structural positioning of m^6^A and its surrounding sequence context (DRACH motif) underlies its selective functional roles (reviewed in [25,26]).

#### 3.1.2. Pseudouridine (Ψ): Stability and Therapeutic Relevance

Pseudouridine (Ψ) is an isomer of uridine in which the uracil base is attached to the ribose via a carbon–carbon instead of a nitrogen–carbon glycosidic bond, increasing base stacking and RNA stability (Figure 1). While Ψ has long been studied in tRNA and rRNA, transcriptome-wide mapping (using Pseudo-seq and CeU-seq) has revealed widespread presence in mRNA, especially under stress conditions [27]. Ψ can be introduced either cotranscriptionally by PUS enzymes or chemically during therapeutic RNA synthesis [28].

Functionally, Ψ enhances mRNA translational capacity and prevents innate immune recognition, which is critical for mRNA therapeutics [29]. In COVID-19 vaccines, synthetic mRNAs are modified with Ψ to reduce Toll-like receptor activation and increase translational efficiency [29]. Ψ also accumulates in stress granules and may contribute to selective mRNA stabilization during oxidative or nutrient stress [30]. Although its functions are still being elucidated, Ψ appears to act as a broad modulator of RNA stability, translation, and immunogenicity [31].

#### 3.1.3. 5-Methylcytosine (m^5^C): An Emerging Modulator of mRNA Fate

The modification 5-methylcytosine (m^5^C) (Figure 1) is catalyzed by the NSUN family of methyltransferases, particularly NSUN2 and NSUN6, and is enriched in both the coding region and 3′ UTR of mRNAs [32]. Unlike m^6^A, m^5^C lacks a well-characterized reader protein repertoire, though ALYREF has been proposed to bind m^5^C-modified mRNA and facilitate nuclear export [32].

Recent studies suggest that m^5^C enhances mRNA stability and influences ribosome loading. NSUN2-mediated m^5^C is upregulated in response to stress and during cell differentiation. Loss-of-function mutations in NSUN2 cause neurodevelopmental disorders, underscoring its importance. In cancer, m^5^C marks are elevated in tumor samples, where they correlate with poor prognosis and resistance to chemotherapy. Although less well-studied than m^6^A, m^5^C represents a growing area of interest in epitranscriptomics due to its context-specific roles in mRNA fate and disease [33,34,35].

Recent evidence reveals extensive crossregulation between the m^6^A and m^5^C RNA methylation systems, including reciprocal modifications of their effector transcripts, suggesting post-transcriptional feedback loops [32]. Proteomic analyses uncovered coordinated cellular responses and interactions—such as between ALKBH5 (m^6^A eraser) and NSUN4 (m^5^C writer)—linked to diverse pathways including mitochondrial function, proteasome activity, and post-translational modifications [32]. These networks also converge on neurological processes, with in vitro data showing colocalization of m^6^A-marked RNAs and the m^5^C reader ALYREF following synaptic activation [32].

#### 3.1.4. Inosine (I): Codon Reprogramming and RNA Editing

Inosine is produced through the enzymatic deamination of adenosine by adenosine deaminases acting on RNA (ADARs) (Figure 1), and it represents one of the most widespread and functionally significant RNA modifications in metazoans [36]. During translation, inosine is interpreted as guanosine by the ribosome, effectively reprogramming codons and enabling the generation of protein isoforms not directly encoded by the genome [36]. This editing commonly occurs in coding regions, 3′ untranslated regions (UTRs), and especially within microRNA (miRNA) seed sequences, where it can drastically alter target specificity [37].

Functionally, A-to-I editing expands transcriptomic and proteomic diversity and plays a critical role in regulating innate immunity, particularly by preventing inappropriate activation of cytosolic RNA sensors such as MDA5 [38]. It also modulates mRNA splicing, nuclear retention, and subcellular localization [39]. In the central nervous system, high levels of editing are essential for the proper function of neurotransmitter receptors and ion channels, such as the GluA2 subunit of AMPA receptors and voltage-gated potassium channels [40].

Dysregulation of A-to-I editing has been implicated in a wide range of diseases, including autoimmune disorders (e.g., Aicardi–Goutières syndrome), neurological diseases (e.g., epilepsy, ALS, schizophrenia), and multiple cancers, where editing imbalances may contribute to immune evasion or altered cell signaling [41,42].

#### 3.1.5. N1-Methyladenosine (m^1^A): Translational Modulator

N1-methyladenosine (m^1^A) is a reversible RNA modification in which a methyl group is added to the N1 position of adenosine (Figure 1), introducing a positive charge at physiological pH [43]. This structural alteration disrupts canonical Watson–Crick base pairing and significantly affects RNA secondary structure, stability, and interaction with ribosomes [43]. The modification m^1^A is commonly enriched in 5′ untranslated regions (5′ UTRs), especially near translation start codons, suggesting a role in fine-tuning translation initiation [44]. It is installed primarily by the methyltransferase complex TRMT6/TRMT61A and removed by the demethylase ALKBH3, indicating tight enzymatic regulation [45].

Functionally, m^1^A can promote or inhibit translation in a position-dependent manner—enhancing initiation when placed proximally to the start codon, but potentially repressing translation when located elsewhere [46]. Its levels are responsive to various cellular stimuli, including nutrient availability, oxidative stress, and heat shock, making m^1^A a dynamic regulator in stress-adaptive gene expression programs [47]. In mitochondria, m^1^A also appears in mitochondrial tRNAs and rRNAs, where it contributes to proper translation of mitochondrial-encoded proteins [48].

Pathologically, elevated m^1^A levels have been observed in multiple cancer types, where they are thought to support oncogenic growth by enabling translational reprogramming toward stress-resilient and proliferation-promoting mRNAs [49]. Overexpression of m^1^A writers such as TRMT6/61A or impaired ALKBH3 activity has been correlated with poor prognosis and chemoresistance [49]. Because of its dynamic nature and disease relevance, m^1^A is emerging as a biomarker for tumor progression and a potential therapeutic target [49].

Technological advances such as m^1^A-selective chemical labeling, antibody-based enrichment (m^1^A-seq), and nanopore-based detection are now enabling more precise, transcriptome-wide mapping of m^1^A at single-nucleotide resolution [50]. Future studies will clarify the tissue-specific and context-dependent roles of m^1^A in development, differentiation, and disease [51].

#### 3.1.6. m^6^Am: Cap-Adjacent Modification

N6,2′-O-dimethyladenosine (m^6^Am) is a methylated nucleotide located immediately adjacent to the canonical 7-methylguanosine (m^7^G) cap at the first transcribed nucleotide of certain mRNAs, typically when the initiating base is adenosine. This dual methylation—on both the N6 position of the adenine base and the 2′-O position of the ribose—creates a structurally distinct cap-proximal mark [52,53]. The modification m^6^Am is deposited by the methyltransferase PCIF1 (also known as CAPAM) in a sequence-dependent manner and removed by the m^6^A demethylase FTO, linking it to the broader dynamic regulation of m^6^A-related pathways [52,53].

Functionally, m^6^Am enhances mRNA transcript stability and has been associated with increased resistance to decapping and exonucleolytic decay, particularly through the DCP2-dependent pathway [52]. As such, it prolongs mRNA half-life and can influence gene expression levels at the post-transcriptional level. Importantly, the presence of m^6^Am at the 5′ end may reduce binding affinity of certain decapping enzymes and modulate translation efficiency under basal or stress conditions [52].

The effects of m^6^Am are context-dependent, varying across tissues, developmental stages, and stress responses [54]. For instance, under hypoxic or nutrient-limiting conditions, dynamic changes in PCIF1 expression or FTO demethylation activity can lead to remodeling of the m^6^Am epitranscriptome, thereby adjusting the stability of transcripts encoding key regulatory proteins. This fine-tuning has been implicated in cellular adaptation to environmental cues [54].

From a disease perspective, altered m^6^Am levels have been observed in cancer, where FTO overexpression may drive oncogenic gene expression by removing m^6^Am and destabilizing tumor suppressor mRNAs [52,53]. Conversely, reduced FTO activity has been linked to metabolic disorders and neurodevelopmental abnormalities [52,53]. However, disentangling the specific contributions of m^6^Am from internal m^6^A sites remains a technical challenge [54].

Recent advances in m^6^Am-specific mapping techniques—such as m^6^Am-seq and cap-specific chemical labeling approaches—are beginning to shed light on the m^6^Am epitranscriptome with greater resolution [54]. Future studies will further elucidate how m^6^Am integrates into broader regulatory networks and how its reversible deposition influences RNA fate in health and disease [52,53].

#### 3.1.7. 5′ Cap Modifications (Cap0, Cap1, Cap2): Orchestrators of Immune Evasion and Translation Control

Eukaryotic mRNAs are cotranscriptionally modified at their 5′ ends with a series of cap structures that play essential roles in RNA stability, nuclear export, translation, and immune recognition [55,56]. These modifications occur in a hierarchical manner and are installed by a coordinated set of enzymes: RNGTT (RNA guanylyltransferase and 5′-phosphatase), RNMT (RNA guanine-N7 methyltransferase), and the cap-specific 2′-O methyltransferases CMTR1 and CMTR2 [57].

The Cap0 structure (m^7^GpppN, where N is any nucleotide) is formed when RNGTT first adds a guanosine cap to the nascent RNA and RNMT methylates the guanosine at the N7 position [57]. This modification is essential for transcript stability and the recruitment of the cap-binding complex (CBC), which facilitates nuclear export and translation initiation [57].

Cap1 and Cap2 modifications involve additional 2′-O-methylation of the ribose sugar of the first and second transcribed nucleotides, respectively. CMTR1 methylates the first nucleotide to form Cap1, and CMTR2 modifies the second nucleotide to yield Cap2 [55,56]. These methylations are crucial for distinguishing endogenous mRNAs from viral RNAs, as innate immune sensors such as RIG-I and MDA5 recognize and respond to improperly capped or unmethylated RNA species [55,56]. Thus, Cap1 and Cap2 modifications act as a molecular “self” signature that prevents erroneous activation of interferon-stimulated gene expression [55,56].

Beyond immune evasion, cap methylation status influences translation efficiency. Cap1 and Cap2 modifications enhance the binding affinity of translation initiation factors (e.g., eIF4E) and are increasingly recognized as modulators of transcript-specific translation under stress or developmental cues. For example, selective capping patterns may allow cells to prioritize translation of certain mRNAs during viral infection, inflammation, or differentiation.

Emerging evidence also suggests dynamic regulation of capping enzymes in cancer, immune disorders, and viral pathogenesis. Some viruses, such as flaviviruses and coronaviruses, encode their own cap-modifying enzymes to mimic host cap structures and evade immune detection.

In summary, 5′ cap modifications (Cap0, Cap1, and Cap2) function not only as mechanical gatekeepers of RNA processing but as immunological checkpoints and regulators of translational prioritization, underscoring their centrality in gene expression regulation and host-pathogen interactions [55,56].

#### 3.1.8. 5-Methyluridine (m^5^U): An Emerging Player in mRNA Regulation

Also referred to as ribothymidine, 5-methyluridine (m^5^U) is a methylated form of uridine where a methyl group is added to the fifth carbon of the uracil ring [58]. Historically, this modification has been best characterized in transfer RNAs (tRNAs), where it plays a well-established role in stabilizing the TΨC loop and facilitating accurate codon-anticodon pairing during translation [58]. In tRNA, m^5^U is catalyzed by enzymes such as TRMT2A and TRMT2B (tRNA methyltransferase 2 homologs A and B), which are conserved across eukaryotes [58].

More recently, attention has turned to the presence and potential functions of m^5^U in messenger RNA (mRNA) and other noncoding RNAs, although its prevalence and biological significance in these contexts remain incompletely understood [59]. Emerging transcriptomic and mass spectrometry evidence suggests that m^5^U may exist in low stoichiometry in certain mRNAs, potentially acting as a regulatory modification under specific physiological or stress conditions [59].

Mechanistically, m^5^U in mRNA may influence local secondary structure, RNA–protein interactions, or ribosome dynamics during translation [59]. It could also affect mRNA stability or splicing, depending on its location within the transcript. For example, methylation within untranslated regions (UTRs) or near splice sites may modulate transcript half-life or alternative exon usage [59].

The enzymatic machinery responsible for m^5^U deposition in mRNA is still under investigation. While TRMT2A/B are known to methylate tRNAs, whether they also modify mRNAs directly—or whether other, yet-uncharacterized RNA methyltransferases are involved—remains an open question [59]. Likewise, demethylases or “erasers” for m^5^U in RNA have not yet been definitively identified, raising the possibility that this mark may be either stable or dynamically regulated through nonenzymatic turnover pathways [59].

The biological significance of m^5^U in mRNA is further underscored by emerging links to cancer biology and neurological diseases, where dysregulation of TRMT2 enzymes and associated pathways has been observed [60]. Additionally, some viral RNAs have been found to incorporate m^5^U, potentially as a mechanism to mimic host RNA and evade immune detection [60].

In summary, although best studied in tRNA, m^5^U is an underexplored but promising RNA modification in mRNA, with potential roles in fine-tuning gene expression, especially under stress or developmental regulation. Future studies using direct RNA sequencing and m^5^U-specific antibodies or chemical labeling methods will be crucial to uncover its distribution, function, and regulatory dynamics in the transcriptome.

#### 3.1.9. 2′-O-Methyladenosine (Am): Emerging Regulatory Roles Beyond tRNA

In the ribonucleotide modification 2′-O-methyladenosine (Am), a methyl group is added to the 2′ hydroxyl (OH) position of the ribose sugar on adenosine (Figure 1). While extensively characterized in transfer RNA (tRNA) and ribosomal RNA (rRNA)—where it contributes to RNA stability, secondary structure, and resistance to nucleolytic cleavage—recent research suggests that Am also appears in messenger RNA (mRNA), particularly near the 5′ end and within untranslated regions (UTRs) [61,62].

In tRNA and rRNA, 2′-O-methylations such as Am help maintain proper folding and enhance resistance to enzymatic degradation. The methylation of adenosine to form Am can prevent base pairing and increase thermal stability, which is particularly important in RNAs exposed to structural fluctuations during processes like translation [63].

The FTSJ1 enzyme (FtsJ RNA 2′-O-methyltransferase 1), a human homolog of the bacterial FtsJ methyltransferase family, is one of the key methyltransferases responsible for installing Am modifications, particularly in tRNA [64]. FTSJ1 has also been implicated in neuronal development, and mutations in this gene are associated with intellectual disability syndromes, suggesting that Am modification may be critical for neurodevelopmental gene regulation [65].

In mRNA, Am has been observed in cap-adjacent positions, where it contributes to the Cap1 and Cap2 structures (see Section 3.1.7), but may also occur internally within transcripts [66]. Cap-associated Am modifications are known to modulate innate immune recognition by reducing the immunogenicity of foreign RNAs and enhancing translational efficiency [66].

Although Am in internal mRNA regions is less well-characterized, emerging evidence points to dynamic regulation under stress conditions and potential roles in alternative splicing, RNA localization, or translational fine-tuning [67]. The presence of Am may influence RNA-binding protein affinity or change the structural dynamics of the mRNA molecule in ways that are still being actively investigated [67].

Am is also thought to contribute to resistance to exonucleolytic degradation, much like other 2′-O-methyl modifications [68]. This may allow Am-containing transcripts to persist longer in the cytoplasm, thereby shaping post-transcriptional gene expression patterns [68].

Overall, 2′-O-methyladenosine represents an evolutionarily conserved but underexplored RNA modification that may play broader roles in the regulation of gene expression, especially under conditions that challenge transcript stability or immune tolerance. Further advances in site-specific detection methods, such as RiboMethSeq [69] or nanopore sequencing [70] approaches adapted for sugar methylation, are expected to shed light on the full extent of Am′s regulatory roles in the transcriptome [61,62].

#### 3.1.10. N4-Acetylcytidine (ac4C): Enhanced Translation Fidelity and Stress Response Modulation

N4-acetylcytidine (ac4C) is a conserved RNA modification in which an acetyl group is added to the amino group at the N4 position of cytidine (Figure 1) [71]. While ac4C was historically studied in tRNA and rRNA—where it contributes to codon–anticodon pairing accuracy and ribosomal stability—recent studies have identified ac4C in messenger RNA (mRNA), particularly in the coding sequences (CDS) of highly expressed genes [71].

The primary “writer” enzyme responsible for installing ac4C in RNA is NAT10 (N-acetyltransferase 10), a protein with both acetyltransferase and helicase domains [72]. NAT10 targets cytidine residues and acetylates them in a sequence- and structure-dependent manner, enhancing the structural integrity of RNA and facilitating accurate base pairing during translation [72]. This modification leads to improved translational fidelity, reducing the frequency of codon misreading or ribosomal stalling [72].

Beyond translation, ac4C also enhances mRNA stability, likely by promoting structural rigidity and protecting transcripts from exonucleolytic decay [73]. This effect may be particularly important under conditions of cellular stress, where global mRNA degradation is elevated, and the selective stabilization of certain transcripts becomes essential for survival [73]. Indeed, ac4C levels are dynamically regulated in response to stress, including oxidative and heat shock conditions, where the modification helps ensure the continued translation of critical survival genes [73].

Intriguingly, elevated levels of ac4C and increased NAT10 expression have been observed in several human cancers, including colorectal and liver cancers [74]. This suggests that ac4C may play a role in tumorigenesis, possibly by enabling the overexpression of oncogenic proteins with greater stability and translational efficiency [74]. Inhibition of NAT10 has been shown to suppress tumor growth in some models, positioning it as a potential therapeutic target [74].

Emerging evidence also links ac4C with the regulation of immune responses and stress granule formation, hinting at broader roles in post-transcriptional control [75]. Moreover, because NAT10 is involved in DNA damage repair and nuclear architecture, its RNA-modifying functions may interface with other cellular pathways beyond translation [75].

In summary, ac4C is a versatile RNA modification that not only enhances translation accuracy and mRNA stability but contributes to cellular adaptation under stress and may promote pathological states such as cancer. As technologies improve for detecting ac4C sites transcriptome-wide (e.g., acRIP-seq and ac4C-specific antibodies), our understanding of its regulatory network and disease relevance is expected to grow significantly [76].

#### 3.1.11. N7-Methylguanosine (m^7^G)

The modification m^7^G forms part of the 5′ cap structure but has also been detected internally in some mRNAs. It plays roles in nuclear export and translation. Recent studies have suggested that mRNA internal m^7^G and its writer protein METTL1 (methyltransferase 1, tRNA methylguanosine) are closely related to cell metabolism and cancer regulation. The IGF2BP (insulin growth factor 2 binding protein)-family proteins IGF2BP1-3 can preferentially bind internal mRNA m^7^G and regulate mRNA stability [77].

#### 3.1.12. 2′-O-Methylguanosine (Gm): RNA Stability and Immune Evasion

In the post-transcriptional RNA modification 2′-O-methylguanosine (Gm), a methyl group is added to the 2′-hydroxyl (2′-OH) group of the ribose sugar in guanosine [78]. This methylation can occur at both internal sites within the RNA molecule and at cap-adjacent positions, particularly within the first few nucleotides of the mRNA 5′ end, such as in Cap1 (m^7^GpppNm) and Cap2 (m^7^GpppNmNm) structures [78].

The enzymatic writers for Gm modifications include members of the FTSJ (FtsJ methyltransferase) family and CMTR2 (Cap Methyltransferase 2) for cap-adjacent Gm. These enzymes catalyze 2′-O-methylation in a sequence- and structure-specific manner, and their activity is often regulated by stress, developmental stage, or cell type [78].

Functionally, Gm contributes to mRNA stability by protecting transcripts from exonucleolytic cleavage. The 2′-O-methyl group enhances the resistance of the phosphodiester backbone to degradation by RNases, especially those that recognize unmodified ribose moieties [78]. This protection against decay increases the half-life of mRNAs and may promote sustained translation of essential proteins, particularly under stress or during rapid proliferation [78].

Additionally, Gm plays a significant role in regulating translation efficiency. By altering RNA secondary structure and reducing backbone flexibility, Gm can enhance ribosome processivity and reduce translational pausing [79]. When located near the 5′ cap, Gm modifications can improve the binding affinity of the eukaryotic initiation factor eIF4E and facilitate the formation of the cap-binding complex, thereby boosting cap-dependent translation initiation [79].

Importantly, 2′-O-methylation also functions as a self-identifying molecular signature that distinguishes endogenous mRNAs from viral or foreign RNAs [80]. The innate immune sensor RIG-I, for instance, is known to preferentially detect RNAs lacking 2′-O-methyl modifications, leading to the activation of antiviral responses [80]. Thus, Gm and related cap-proximal modifications help evade immune detection, maintaining self-tolerance and preventing unwanted inflammation [80].

Emerging evidence suggests that dysregulation of Gm deposition or recognition may contribute to diseases such as cancer and neurodegeneration, where aberrant translation and immune activation are commonly observed [81]. However, the specific distribution and functional significance of internal Gm residues in mRNA are still being actively explored with the help of high-resolution techniques like RiboMeth-seq and LC-MS/MS [82,83].

In summary, 2′-O-methylguanosine (Gm) is a versatile RNA modification that enhances transcript stability, improves translational efficiency, and contributes to immune self-recognition—underscoring its critical role in both normal physiology and disease [84].

#### 3.1.13. 2′-O-Methylcytidine (Cm): RNA Stability, Translation, and Immune Modulation

In the ribose modification 2′-O-methylcytidine (Cm), a methyl group is added to the 2′-hydroxyl (2′-OH) group of cytidine (Figure 1) [78]. Like other 2′-O-methylated nucleotides, Cm is found in both internal sites of the RNA body and at cap-adjacent positions as part of higher-order cap structures (Cap1 and Cap2) [78]. It is deposited by methyltransferases such as FTSJ1 and CMTR1/2, often in a sequence- and structure-dependent manner [78].

Functionally, Cm contributes to mRNA stability by enhancing resistance to RNase-mediated degradation. The methylation at the 2′-OH position sterically hinders endonucleolytic cleavage, thereby prolonging transcript half-life and enabling sustained protein production—especially important under conditions of cellular stress or immune challenge [68,85].

Cm also plays a critical role in modulating translation efficiency. By rigidifying the ribose–phosphate backbone and altering local RNA secondary structure, it can promote more efficient ribosomal scanning and elongation [68,85]. When located near the 5′ cap, Cm may also facilitate the recruitment of initiation factors such as eIF4E and eIF4G, enhancing cap-dependent translation initiation [68,85]. This may be especially important in highly proliferative or metabolically active cells where translational throughput is critical.

In the innate immune system, Cm and other 2′-O-methylations serve as self-marks that prevent activation of immune sensors such as RIG-I and MDA5, which preferentially detect hypomethylated or nonself RNA [86]. This capacity for immune evasion has been exploited by some viruses, which encode their own 2′-O-methyltransferases to mimic host mRNAs [86]. Dysregulation of Cm methylation may therefore contribute to autoimmune disorders or altered antiviral defenses [86].

Though less studied than other modifications such as m^6^A, emerging evidence from RiboMeth-seq, LC-MS/MS, and Nanopore direct RNA sequencing indicates that Cm occurs at conserved positions in both coding and noncoding RNAs [87]. Its distribution may vary by cell type and environmental context, and changes in Cm levels have been linked to cancer progression and neurological disorders [88].

In summary, 2′-O-methylcytidine (Cm) serves as a multifunctional RNA modification that enhances mRNA stability, supports efficient translation, and helps maintain immune homeostasis. Its expanding functional repertoire underscores the complexity of the epitranscriptome and the need for further high-resolution mapping and functional studies [84].

#### 3.1.14. 5-Hydroxymethylcytidine (hm^5^C): Potential Epitranscriptomic Regulator with Epigenetic Parallels

A modified cytidine residue, 5-hydroxymethylcytidine (hm^5^C) is characterized by a hydroxymethyl group (-CH_2_OH) at the 5-position of the pyrimidine ring [89]. Although well-characterized in the context of DNA as an intermediate in the active demethylation of 5-methylcytosine (5mC) by TET (ten–eleven translocation) enzymes, the presence of hm^5^C in RNA suggests a potential parallel regulatory role in the epitranscriptome [89]. Its discovery in multiple RNA classes—including mRNA, rRNA, and tRNA—has raised important questions about its biogenesis, function, and dynamics in gene expression regulation [90].

In RNA, hm^5^C is likely derived from the oxidation of 5-methylcytidine (m^5^C) via TET family dioxygenases, particularly TET2, although the full set of responsible enzymes and their cofactor requirements (e.g., Fe^2+^, α-ketoglutarate) remain to be fully elucidated [90]. The modification may be transient, serving as a signaling intermediate or as a stable modification depending on context and cell type [90].

Functionally, hm^5^C is hypothesized to influence RNA structure, binding protein recruitment, and post-transcriptional regulation, much like its m^5^C precursor. Its enrichment in coding sequences and untranslated regions suggests possible involvement in mRNA stability, translation regulation, or nuclear export [90], although these functions remain speculative in the absence of comprehensive mechanistic studies.

The epitranscriptomic role of hm^5^C may also mirror its epigenetic behavior in DNA, wherein it acts as both a stable regulatory mark and a reversible intermediate during demethylation. This raises the possibility of dynamic RNA demethylation cycles, potentially responsive to environmental stimuli, cellular stress, or developmental cues [90]. In support of this, emerging studies have found context-dependent patterns of hm^5^C in mammalian tissues, including neurons, and in disease states such as cancer and neurodegeneration [90].

Recent advances in mapping technologies such as chemical labeling, antibody-based enrichment, and direct RNA nanopore sequencing have enabled more confident detection of hm^5^C sites in the transcriptome [91]. However, its stoichiometry and biological significance remain poorly defined, in part because of a lack of selective reader proteins and functional assays [91].

In conclusion, while hm^5^C is a relatively understudied RNA modification, its structural similarity and biosynthetic overlap with DNA methylation pathways suggest it may represent a novel layer of epitranscriptomic regulation, potentially contributing to cell state plasticity, translational control, and disease pathogenesis. Further studies are warranted to clarify its readers, erasers, and downstream functional consequences (reviewed in [33]).

#### 3.1.15. Comodified m^6^A/Ψ Sites: Combinatorial Control of RNA Fate

The discovery of comodified RNA loci, particularly those harboring both N6-methyladenosine (m^6^A) and pseudouridine (Ψ) modifications, represents an emerging frontier in the field of epitranscriptomics [92]. These co-occurring marks, located on the same RNA molecule and sometimes within the same nucleotide neighborhood, suggest a layer of combinatorial regulation that mirrors the complexity of histone modification crosstalk in chromatin biology [92]. While m^6^A is known to influence RNA splicing, export, and decay through reader proteins such as YTH domain-containing proteins, Ψ contributes to RNA stability and secondary structure by enhancing base stacking and altering hydrogen bonding potential [92].

The presence of m^6^A and Ψ at adjacent or overlapping sites raises key questions about how these modifications interact structurally and functionally [92]. It is possible that m^6^A-induced conformational changes may prime an RNA region for subsequent pseudouridylation, or vice versa, and that together they cooperatively modulate translation efficiency, RNA–protein interactions, or decay rates. For instance, m^6^A may promote recruitment of RNA-binding proteins while Ψ stabilizes the structure of the binding motif, fine-tuning accessibility and recognition in a context-dependent manner.

Long-read nanopore direct RNA sequencing has emerged as a powerful tool for detecting comodified sites because of its ability to read full-length RNA molecules while preserving native modifications [92]. This allows for precise mapping of modification co-occurrence on individual transcripts and even single molecules—something not easily achievable with traditional short-read or antibody-based approaches. Studies using this technology have already revealed previously unrecognized modification patterns in stress-responsive and developmentally regulated mRNAs [8].

Functionally, comodified sites may play roles in adaptive gene expression, particularly under stress conditions or during development when dynamic remodeling of RNA fate is essential. Preliminary findings suggest that dual m^6^A/Ψ marks are enriched in mRNAs involved in cell fate decisions, metabolic control, and immune signaling, though mechanistic details remain under investigation [92].

In sum, comodified m^6^A/Ψ loci represent a nascent but promising area of study that expands our understanding of multilayered RNA regulation. These sites may act as epitranscriptomic integration hubs, where multiple signals converge to direct transcript-specific outcomes. As tools for multimodification mapping improve, future research will be crucial for uncovering how comodified RNAs are written, read, erased, and functionally interpreted in health and disease [8].

### 3.2. Positional Enrichment of mRNA Modifications

RNA modifications are not randomly distributed across transcripts; rather, they exhibit specific positional enrichment patterns that are tightly linked to their regulatory functions (Table 1).

▪m^6^A: Highly enriched near stop codons and within 3′ UTRs. This spatial positioning facilitates regulated decay and translational control via reader proteins such as YTHDF2 [93].▪▪ Ψ: Broadly distributed across coding sequences and UTRs. Stress-induced Ψ sites often appear in transcripts involved in stress response and cancer [94].▪m^5^C: Localized to coding regions and 3′ UTRs. m^5^C sites tend to enhance stability and promote nuclear export [35].▪Inosine (I): Common in coding regions and 3′ UTRs, where it arises from ADAR-mediated A-to-I editing. Inosine affects codon identity, splicing, and miRNA targeting [95].▪m^1^A: Found near start codons and 5′ UTRs. It can either promote or repress translation depending on its exact position and structural context [27].▪m^6^Am: Located adjacent to the 5′ cap, m^6^Am increases mRNA stability and is deposited by the PCIF1 methyltransferase [96].▪ac4C: Predominantly enriched in coding regions of highly translated genes, where it enhances translation fidelity [71].

These positional signatures reflect the coevolution of epitranscriptomic marks with RNA-binding proteins and cellular pathways. Emerging technologies such as nanopore direct RNA sequencing and site-specific chemical labeling continue to refine our understanding of these spatial patterns and their functional consequences.

### 3.3. Interpretation of Modification Ranking

The prevalence of publications reflects not only biological abundance but technical detectability and perceived functional importance. m^6^A dominates the field because of the early availability of high-affinity antibodies and the development of m^6^A-seq, which catalyzed mechanistic discoveries across diverse biological systems [5,97]. Pseudouridine and m^5^C followed as sequencing and chemical mapping methods improved [31,98].

Inosine ranks highly because of its role in transcriptome diversification through RNA editing—a uniquely dynamic modification that alters coding potential. The high rank of cap modifications highlights their long-known essential role in translation and immune modulation, particularly relevant to viral and vaccine RNA biology. ICE-seq (Inosine chemical erasing) was developed in 2015, which helps explain the high ranking of inosine in mRNA publications [99].

Lower-ranked modifications such as ac4C, Gm, Cm, and hm^5^C likely suffer from limited detection methods and ambiguous biological roles rather than true scarcity. The low ranking of m^6^A/Ψ comodifications underscores how technical limitations may obscure complex regulatory interplay, which future single-molecule and multimodification sequencing technologies are poised to reveal [8].

### 3.4. Disease Relevance of Top RNA Modifications

RNA modifications play emerging roles in development, disease, and therapy. The m^6^A mRNA modification is implicated in cancer progression [100], stem cell differentiation [101], and neurodevelopmental disorders [102]. Its dysregulation via altered METTL3 or FTO expression is linked to glioblastoma [103], leukemia [104], and metabolic diseases [105].

Pseudouridine is foundational in the design of synthetic mRNA therapeutics—particularly COVID-19 vaccines—which use Ψ to evade innate immune detection and enhance translation [29]. The m^5^C RNA modifications or NSUN2 deletions are associated with intellectual disability [106,107] and cancer [108].

Inosine levels are altered in neurodegenerative diseases such as ALS [109] and in immune dysregulation syndromes [14,110]. m^1^A and m^6^Am modulate translation efficiency and stress responses [111], potentially contributing to cancer cell plasticity and adaptation to hypoxia [112].

Emerging modifications (e.g., ac4C, hm^5^C) may become important biomarkers or therapeutic targets as detection tools improve [113]. Understanding how these marks influence gene expression in disease-relevant tissues remains a major unmet need.

The growing relevance of epitranscriptomic alterations in human disease underscores the importance of developing high-throughput, base-specific detection tools to enable functional and diagnostic studies across a broad spectrum of RNA modifications.

**Table 1 genes-16-00951-t001:** Most well-known mRNA modifications ranked in order of PubMed citations.

Rank ^1^	Modification	Abbreviation	Enzyme(s)	Role	Positional Enrichment	~N cited	Reference(s)
1	N6-methyladenosine	m^6^A	METTL3, METTL14, FTO, ALKBH5	Splicing, translation, decay, export	Near stop codon and 3′ UTR	>7000	[114,115,116]
2	Pseudouridine	Ψ	PUS1, PUS7	Stability, decoding, stress response	Internal coding region	~1000	[117,118]
3	5-methylcytidine	m^5^C	NSUN2, DNMT2	Export, stability	3′ UTR	~800	[119,120]
4	Inosine	I	ADAR1, ADAR2	A-to-I editing, recoding	dsRNA regions	~750	[121]
5	N1-methyladenosine	m^1^A	TRMT6/TRMT61A	Translation initiation, structure	5′ UTR near start codon	~400	[122,123]
6	N6,2′-O-dimethyladenosine	m^6^Am	PCIF1	Cap-proximal stability	+1 position after 5′ cap	~200	[124]
7	5′ cap modifications	Cap0, Cap1, Cap2	RNGTT, RNMT, CMTR1, CMTR2	Immune evasion, translation	5′ cap	~150–300	[125,126]
8	5-methyluridine	m5U	TRMT2A/B	tRNA-like stability role in mRNA	tRNA mimic sites	<100	[127]
9	2′-O-methyladenosine	Am	FTSJ1, CMTR1	Cap stability and processing	Near 5′ cap	<100	[126,128]
10	N4-acetylcytidine	ac4C	NAT10	Translation, stress response	Internal coding region	<50	[129]
11	N7-methylguanosine	m^7^G	RNGTT, RNMT, METTL1	5′ cap structure, nuclear export	5′ cap	<50	[125,130]
12	2′-O-methylguanosine	Gm	CMTR2	Cap and internal stability	5′ cap	<30	[125,126]
13	2′-O-methylcytidine	Cm	FTSJ1	Stability, cap modification	5′ cap proximal	<30	[62]
14	5-hydroxymethylcytidine	hm^5^C	TET2	Epigenetic-like regulation	3′ UTR	<20	[131]
15	m^6^A:Ψ comodified sites	m^6^A/Ψ	Multiple	Dynamic regulation, RNA structure	Near stop codon, 3′ UTR	<10	[131]

^1^ Rank is based on the number of PubMed citations (see Materials and Methods). References are the first or most relevant PubMed citation of the enzymes involved in the mRNA modification.

## 4. Discussion

The findings presented in this review emphasize the central role of m^6^A in mRNA regulation, reflecting both its biological relevance and the technical accessibility of its detection. The strong representation of m^6^A in the literature—surpassing 7000 PubMed citations—demonstrates its position as a dominant epitranscriptomic regulator involved in nearly every stage of mRNA life, from splicing and nuclear export to translation and degradation (reviewed in [25]). This prominence has been further amplified by the development of antibody-based enrichment techniques and transcriptome-wide mapping methods such as MeRIP-seq, which have made m^6^A one of the most tractable RNA modifications for large-scale studies [132].

In contrast, other modifications such as pseudouridine (Ψ), m^5^C, and inosine also show significant presence in the literature, but their detection and interpretation have historically required more specialized tools. Ψ, for instance, has been associated with enhanced transcript stability and stress response yet lacks reliable transcriptome-wide detection without specialized chemical treatment [31]. Similarly, A-to-I editing by ADARs, while essential for neural and immune development, is sometimes overlooked in the context of dynamic mRNA regulation (reviewed in [133]).

RNA capping is a critical post-transcriptional modification that governs RNA stability, processing, and translational efficiency. The canonical eukaryotic cap structures—Cap0 (m^7^GpppN), Cap1 (m^7^GpppNm), and Cap2 (m^7^GpppNmNm)—play essential roles in promoting translation initiation and protecting transcripts from innate immune surveillance. Despite their fundamental importance, these cap modifications remain underrepresented in the literature, likely because of longstanding technical challenges in isolating and sequencing intact 5′ caps, especially within full-length mRNAs. Recent advances, however, have begun to reveal the broader biological relevance of cap methylation patterns. Differential cap modifications are now recognized as key determinants in viral mimicry, host–pathogen interactions, and the optimization of synthetic mRNA therapeutics. The field has also undergone a paradigm shift with the discovery of noncanonical RNA caps, initially in bacteria and now recognized across all domains of life. The repertoire of RNA caps has expanded well beyond the classic m^7^G structure to include metabolite-derived caps such as NAD^+^, FAD, coenzyme A (CoA), UDP-glucose, and ADP-ribose. In addition, cells produce dinucleoside polyphosphate “alarmone” caps and methylated phosphate-containing cap-like structures. These noncanonical caps open new avenues for studying RNA regulation, signaling, and host–pathogen dynamics while also posing intriguing questions about cap recognition and processing machinery across species. (reviewed in [134]).

The relatively low citation counts for certain mRNA modifications—such as N^1^-methyladenosine (m^1^A), 5-methyluridine (m^5^U), and 2′-O-methyladenosine (Am)—should not be interpreted as evidence of limited biological significance. Instead, they reflect persistent technological and methodological barriers that have hindered our ability to detect, map, and quantify these modifications at transcriptome-wide scale in messenger RNAs. In contrast, these marks have long been studied in the context of tRNA and rRNA, where they are more abundant and their structural or functional roles more clearly defined. For example, m^1^A is known to alter base pairing and RNA secondary structure, which can influence translation initiation or pause sites (reviewed in [135]), yet its detection in mRNA is complicated by its lability and the need for specialized chemical treatment or ultrasensitive sequencing approaches. Similarly, m^5^U and Am, though functionally well-established in noncoding RNAs and tRNA fragments [136], lack high-throughput mapping tools specific to their low abundance and dispersed presence in mRNAs. Many of these modifications require enrichment strategies or site-specific antibodies that are currently unavailable or unreliable.

These technical limitations result in underrepresentation of such modifications in both experimental studies and the literature. As novel chemical labeling methods, direct RNA sequencing platforms (e.g., nanopore), and improved base calling algorithms evolve, the field is poised to uncover the broader regulatory roles these modifications may play in mRNA metabolism, localization, and translation—particularly under stress or in specialized cell types. Thus, low citation prevalence is more a reflection of our current technological blind spots than an accurate measure of functional importance.

These findings underscore the urgent need for comprehensive and unbiased platforms capable of detecting the full spectrum of RNA modifications. Nanopore-based direct RNA sequencing—elaborated in the Future Directions section—offers a particularly promising solution. Unlike traditional methods that rely on reverse transcription, nanopore sequencing reads native RNA molecules directly, enabling the detection of base-specific modifications through characteristic shifts in ionic current. A major advantage of this approach is its ability to simultaneously detect multiple modifications along a single full-length mRNA transcript, preserving the contextual relationships between marks [136]. As the accuracy, resolution, and computational interpretation of nanopore data continue to improve, this technology has the potential to transform the field—broadening our understanding of RNA modifications across diverse transcript classes, cell types, and physiological conditions.

## 5. Conclusions and Future Directions

The field of RNA modification research is rapidly evolving, yet significant challenges remain in the comprehensive identification and functional annotation of the full diversity of RNA modifications. The MODOMICS database currently catalogs over 335 distinct chemical modifications of RNA, most of which remain poorly characterized in terms of their distribution, dynamics, and biological function (Figure 2) [4]. A central future goal is the development of both biochemical and sequencing technologies capable of detecting these modifications with high sensitivity and specificity. Among the most promising approaches are antibody-based enrichment techniques and direct RNA sequencing using nanopore technologies. However, the current suite of available antibodies targets only a limited set of well-studied modifications such as m^6^A, m^5^C, and pseudouridine [137,138]. There is a pressing need to develop a broader repertoire of high-affinity, modification-specific antibodies, ideally with minimal crossreactivity, to facilitate immunoprecipitation-based enrichment and mapping of additional modification types across the transcriptome [138].

Parallel to biochemical methods, advances in nanopore sequencing hold transformative potential for detecting RNA modifications directly, without prior conversion or enrichment. The Oxford Nanopore platform has already demonstrated the ability to detect some common modifications by characteristic shifts in ionic current (reviewed in [139]), but most of the 335 known modifications in the MODOMICS database remain indistinguishable because of limitations in both experimental resolution and current basecalling algorithms [6]. Future work must therefore focus on enhancing both the hardware sensitivity of nanopores and the machine-learning-based bioinformatics tools that interpret the raw electrical signal [4]. These improvements will require training datasets that include synthetic RNAs bearing single, known modifications in defined sequence contexts [140]. Ideally, comprehensive modification “barcodes” could be integrated into public databases to facilitate community-wide benchmarking [6].

The MODOMICS database assigns a unique one-character code to each of the more than 500 known natural and synthetic RNA modifications, enabling streamlined annotation and computational analysis [6]. To accommodate the growing diversity of chemical modifications beyond the canonical nucleotides (A, U, G, C), MODOMICS utilizes an extended character set that includes not only Latin letters but symbols from Greek, Cyrillic, and Chinese scripts, as well as other Unicode characters. This expansive coding system allows for the efficient representation of structurally diverse nucleotide variants, including rare, heavily modified, or synthetic analogs used in RNA therapeutics and research [6]. While powerful, this multilingual symbolic system poses challenges for data standardization, software compatibility, and crossplatform interoperability—highlighting the need for harmonized bioinformatics tools capable of parsing and visualizing these expanded alphabets within RNA sequences.

In conclusion, future directions in RNA modification research must encompass expansion of antibody and nanopore-based technologies to detect the full spectrum of natural modifications in the context of evolutionary biology.

## Figures and Tables

**Figure 1 genes-16-00951-f001:**
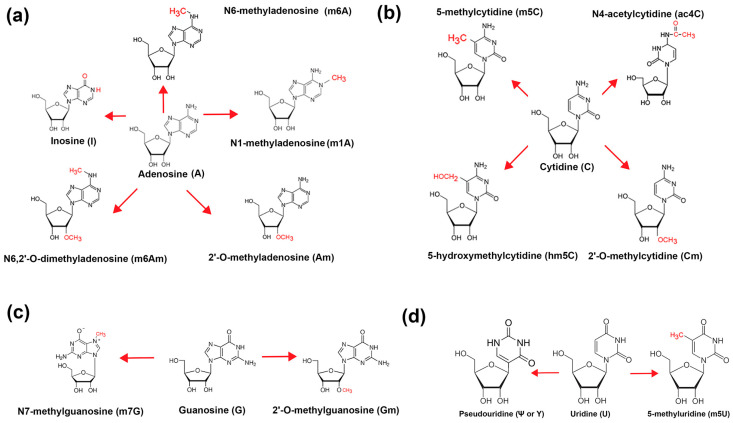
Modifications of bases on mRNA. (**a**) Adenosine modifications; (**b**) cytidine modifications; (**c**) guanosine modifications; and (**d**) uridine modifications.

**Figure 2 genes-16-00951-f002:**
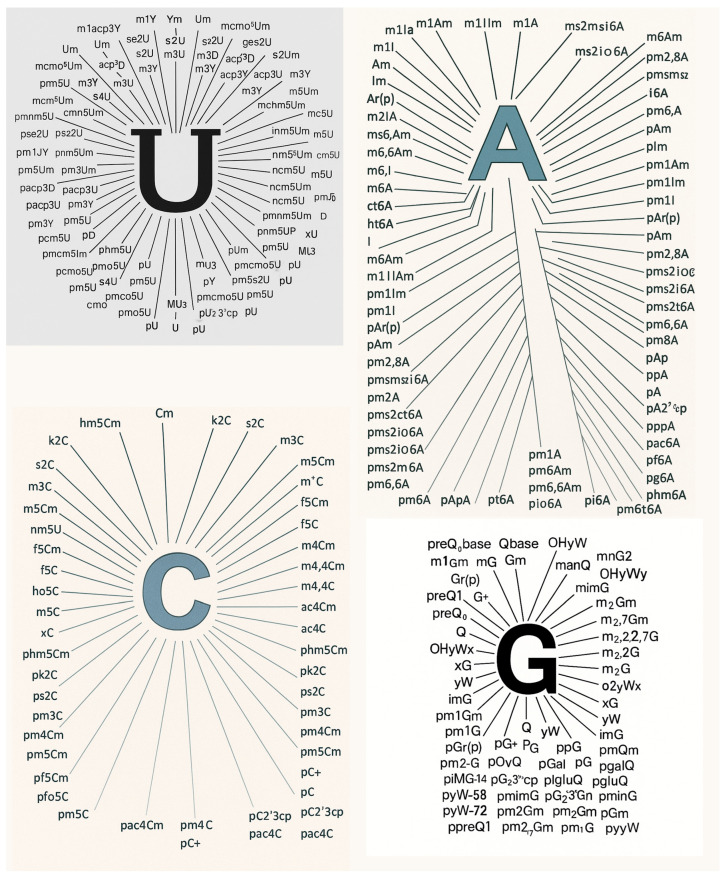
The 335 naturally occurring nucleosides and nucleotides in the most recent MODOMICS database, with their distribution shown on U, A, C, and G.

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
