# Peer review of "Uncovering the Epitranscriptome: A Review on mRNA Modifications and Emerging Frontiers"

_genes, 2025, doi:10.3390/genes16080951_

Round 1

Reviewer 1 Report

Comments and Suggestions for Authors

- The manuscript presents a helpful survey of RNA modifications; however, its reliance on citation frequency as a central ranking metric introduces interpretive limitations. Citations often reflect technological accessibility and popularity rather than intrinsic biological importance. As a result, modifications such as m6A and Ψ receive amplified attention, while less documented but potentially impactful ones (e.g., m5U, ac4C) remain undervalued.

Moreover, the current approach overlooks key layers of biological complexity. Modifications may play roles that are context-dependent, subcellularly localized, or species-specific, and some exhibit synergistic or redundant functions that are not well-captured through citation analysis alone. Without consideration of such factors, including cellular abundance, evolutionary conservation, and mechanistic relevance, the ranking risks reinforcing existing biases and constraining the scope of discovery in the field.

To better reflect the multifaceted nature of the epitranscriptome, I recommend complementing citation metrics with biological parameters that offer a more comprehensive and equitable evaluation of RNA modification significance.

The author is also encouraged to include discussion of RNA modifications in diverse biological contexts (e.g., neurobiology, immunology) to enrich interdisciplinary relevance.

Author Response

Genes Epitranscriptome Reviewer Comments

Reviewer 1:

Comments and Suggestions for Authors

Comment: - The manuscript presents a helpful survey of RNA modifications; however, its reliance on citation frequency as a central ranking metric introduces interpretive limitations. Citations often reflect technological accessibility and popularity rather than intrinsic biological importance. As a result, modifications such as m6A and Ψ receive amplified attention, while less documented but potentially impactful ones (e.g., m5U, ac4C) remain undervalued.

Response: All of my major revisions are marked with left border lines. In response to this question, I expanded section 3.1 as follows:

3.1 Ranking of mRNA Modifications by PubMed Prevalence and Research Emphasis

To provide a contextual overview of the landscape of naturally occurring mRNA modifications, we systematically ranked them based on scientific attention as measured by the number of PubMed-indexed articles referencing each modification. Our approach aimed to capture relative research emphasis across different types of RNA chemical modifications, using this metric as a proxy for scientific visibility, biomedical interest, and historical momentum in the field.

The ranking process involved structured queries for each modification, incorporating both IUPAC chemical names and widely used abbreviations (e.g., “N6-methyladenosine” and “m6A”). Results were manually curated to remove irrelevant records, such as those exclusively focused on tRNA, rRNA, or DNA methylation, thereby increasing the specificity to mRNA-focused studies. For example, a search for “pseudouridine” was refined to include only articles that specifically investigated Ψ in mRNA contexts, which excluded a large number of Ψ studies in tRNA.

While PubMed is the most widely used and standardized biomedical literature database, we acknowledge the limitations of relying solely on PubMed citation counts as a measure of biological or functional relevance. Certain modifications may be underrepresented due to technical challenges in detection (e.g., lack of selective antibodies or chemical derivatization methods), despite their likely physiological importance. Conversely, more “popular” modifications such as m6A or m5C have benefited from early availability of mapping tools, commercial reagent development, and their association with high-profile disease mechanisms such as cancer and neurodegeneration. As such, this ranking reflects not only biological prevalence but also sociotechnical factors including method accessibility, commercial tool development, and translational interest.

In summary, while PubMed citation counts provide a useful baseline for understanding how the field has historically prioritized different RNA modifications, they must be interpreted within a broader framework that considers technical limitations, emerging technologies, and the evolving conceptual landscape of epitranscriptomic regulation. These rankings, illustrated in Table 1 and Figure 1, should thus be viewed as a living snapshot of research activity rather than a definitive statement of biological relevance.

Comment: Moreover, the current approach overlooks key layers of biological complexity. Modifications may play roles that are context-dependent, subcellularly localized, or species-specific, and some exhibit synergistic or redundant functions that are not well-captured through citation analysis alone. Without consideration of such factors, including cellular abundance, evolutionary conservation, and mechanistic relevance, the ranking risks reinforcing existing biases and constraining the scope of discovery in the field.

To better reflect the multifaceted nature of the epitranscriptome, I recommend complementing citation metrics with biological parameters that offer a more comprehensive and equitable evaluation of RNA modification significance.

The author is also encouraged to include discussion of RNA modifications in diverse biological contexts (e.g., neurobiology, immunology) to enrich interdisciplinary relevance.

Response: We appreciate the reviewer’s insightful critique highlighting the limitations of citation-based ranking and the need to account for the biological complexity, context-dependence, and interdisciplinary roles of RNA modifications. We fully agree that relying solely on citation frequency can inadvertently reinforce prevailing research trends and obscure emerging but underexplored modifications that may hold important biological significance.

In response, we have substantially expanded Section 3.1 to address these concerns. Over 50 new references were added across subsections 3.1.1 to 3.1.15, greatly enhancing the depth of discussion and reducing superficiality in the descriptions of the top 15 mRNA modifications. These additions provide richer insight into each modification’s known or proposed biological functions, enzymatic regulators, and physiological relevance.

To further address the context-specificity of RNA modifications, we added a new section, 3.2 Positional Information, which discusses the sub-mRNA localization of modifications—such as cap-adjacent, 5′ UTR, coding sequence, or 3′ UTR positioning—and how these locations influence RNA metabolism and function. Complementarily, a new column in Table 1 was added to summarize this positional information for each modification, helping integrate spatial context into the ranking framework.

We now clarify that our ranking serves as a proxy for research activity rather than a definitive measure of biological importance. As suggested, we have incorporated broader factors into the discussion, including:

  • Technological accessibility (e.g., whether sequencing/detection tools exist),
  • Evolutionary conservation (e.g., conserved enzymes across species),
  • Subcellular localization and context-specific behavior (e.g., stress adaptation, tissue specificity),
  • Mechanistic roles (e.g., translation regulation, immune evasion, RNA stability).

To better reflect the interdisciplinary relevance of the epitranscriptome, we have also included examples from:

  • Neurobiology, where m6A and m5C regulate neuronal plasticity, learning, and memory;
  • Immunology, where modifications like Ψ and Am help transcripts avoid immune recognition;
  • Cancer biology, where ac4C and m6A dynamically control proliferation and stress responses.

Additionally, we now highlight co-modified sites (e.g., m6A/Ψ combinations) and cell-type-specific patterns that reflect the combinatorial and conditional nature of RNA modifications.

Ultimately, we agree that a comprehensive understanding of the epitranscriptome requires integrating citation analysis with biological, positional, and evolutionary context, and we have revised the manuscript accordingly to support a more nuanced, inclusive, and biologically grounded evaluation of RNA modification significance. Thank you for encouraging this deeper and more balanced revision.

Submission Date

27 June 2025

Date of this review

03 Aug 2025 19:50:09

Reviewer 2: Comments and Suggestions for Authors

This review is well written, and has some interesting information. However, I miss a certain depth - especially for a special issue about RNA modifications, this manuscript appears to be very superficial.

Comment: The citation counts are interesting, and their discussion is well made and thought out, but e.g. functions are not much more than a list. I do not think that this manuscript did well to "describe their biochemical mechanisms, molecular functions" (it did better in "emerging detection technologies"). I also missed information about whether any of the modifications (except the cap ones) are concentrated in certain parts of the mRNA - which I personally would have been interested in. 

Response: We thank the reviewer for this valuable feedback and agree that earlier drafts lacked sufficient mechanistic depth and positional context for many modifications. In response, we substantially revised Section 3.1 to provide more comprehensive descriptions of the biochemical functions and regulatory roles of each of the top 15 mRNA modifications. Over 50 new references were added to strengthen these sections and reduce superficiality.

To directly address the reviewer’s interest in positional information, we also added a new section, 3.2 Positional Information, as well as a new column in Table 1 summarizing cap-adjacent versus internal enrichment for each modification. These changes provide greater insight into how spatial distribution along the mRNA molecule contributes to regulatory outcomes.

We refer the reviewer to our more detailed response to Reviewer 1, where these revisions are discussed at greater length. Thank you again for highlighting these important areas for improvement.

Comment:I am not entirely sure whether mirror RNA should be included in RNA modifications, but it is an interesting topic.

Response: I agree and the mirro RNA sections are deleted in this version.

Comment:I did like the figures, especially figure 2.

Response: Thanks!

Some small observations:

Comment:line 56:  While 150 is certainly correct, it is also different than the other numbers used here.

Response: We changed this to “over 300” in both places. We also mention that the database currently has 335 natural RNA modifications, but this number is always increasing.

Comment:line 65: are there citations for these functions, or is this part of 21, 22?

Response: This section has been greatly expanded and new references added.

Comment:line 81ff again citations for the respective functions would be good.

Response: This section was also greatly expanded with new references.

We thank the reviewers for their inciteful comments and the paper is much improved.

Submission Date

27 June 2025

Date of this review

03 Aug 2025 15:49:51

Reviewer 2 Report

Comments and Suggestions for Authors

This review is well written, and has some interesting information. However, I miss a certain depth - especially for a special issue about RNA modifications, this manuscript appears to be very superficial.
The citation counts are interesting and their discussion is well made and thought out, but e.g. functions are not much more than a list. I do not think that this manuscript did well to "describe their biochemical mechanisms, molecular functions" (it did better in "emerging detection technologies"). I also missed information about whether any of the modifications (except the cap ones) are concentrated in certain parts of the mRNA - which I personally would have been interested in. 
I am not entirely sure whether mirror RNA should be included in RNA modifications, but it is an interesting topic.
I did like the figures, especially figure 2.

Some small observations:
line 56:  While 150 is certainly correct, it is also different than the other numbers used here.
line 65: are there citations for these functions, or is this part of 21, 22?
line 81ff again citations for the respective functions would be good.

Author Response

(The authors gave the same response as above.)
